# Dipolar Glass Polymers Containing Polarizable Groups as Dielectric Materials for Energy Storage Applications. A Minireview

**DOI:** 10.3390/polym11020317

**Published:** 2019-02-13

**Authors:** Sebastián Bonardd, Viviana Moreno-Serna, Galder Kortaberria, David Díaz Díaz, Angel Leiva, César Saldías

**Affiliations:** 1Departamento de Química Física, Facultad de Química y de Farmacia, Pontificia Universidad Católica de Chile, Macul, Santiago 7820436, Chile; s.bonardd.salvador@gmail.com (S.B.); vmorenoserna@gmail.com (V.M.-S.); 2Universidad País Vasco/Euskal Herriko Unibertsitatea, “Materials + Technologies” Group, Dpto. Ingeniería Química y Medio Ambiente, Escuela Univ Politécnica, Pza Europa 1, 20018 Donostia-San Sebastián, Spain; galder.cortaberria@upv.eus; 3Institut für Organische Chemie, Universität Regensburg, Universitätsstr. 31, 93053 Regensburg, Germany; david.diaz@chemie.uni-regensburg.de; 4Instituto de Productos Naturales y Agrobiología del CSIC, Avda. Astrofísico Francisco Sánchez 3, 38206 La Laguna, Tenerife, Spain

**Keywords:** dipolar glass polymer, dielectric materials, glass transition temperature

## Abstract

Materials that have high dielectric constants, high energy densities and minimum dielectric losses are highly desirable for use in capacitor devices. In this sense, polymers and polymer blends have several advantages over inorganic and composite materials, such as their flexibilities, high breakdown strengths, and low dielectric losses. Moreover, the dielectric performance of a polymer depends strongly on its electronic, atomic, dipolar, ionic, and interfacial polarizations. For these reasons, chemical modification and the introduction of specific functional groups (e.g., F, CN and R−S(=O)_2_−R´) would improve the dielectric properties, e.g., by varying the dipolar polarization. These functional groups have been demonstrated to have large dipole moments. In this way, a high orientational polarization in the polymer can be achieved. However, the decrease in the polarization due to dielectric dissipation and the frequency dependency of the polarization are challenging tasks to date. Polymers with high glass transition temperatures (T_g_) that contain permanent dipoles can help to reduce dielectric losses due to conduction phenomena related to ionic mechanisms. Additionally, sub-Tg transitions (e.g., γ and β relaxations) attributed to the free rotational motions of the dipolar entities would increase the polarization of the material, resulting in polymers with high dielectric constants and, hopefully, dielectric losses that are as low as possible. Thus, polymer materials with high glass transition temperatures and considerable contributions from the dipolar polarization mechanisms of sub-T_g_ transitions are known as “dipolar glass polymers”. Considering this, the main aspects of this combined strategy and the future prospects of these types of material were discussed.

## 1. Introduction

Recently, the development of innovative materials with outstanding dielectric properties has attracted much attention due to their potential applications in energy storage, digital memory devices, pulsed power systems, and signal processing. Mathematically, the stored energy, W, in a capacitor device can be expressed as:(1)$$W=\frac{1}{Ad}\int^{\;}V\;dq=\int^{\;}E\;dD$$
where *V* is the applied voltage across a capacitor with area *A*, *d* is the dielectric thickness, *E* is the electric field, *D* is the electric displacement, and *q* is the free surface charge. Considering a linear dielectric material, the electric displacement is given by:(2)$$D=\varepsilon_{0}\varepsilon_{T}E$$

Replacing in Equation (1), the following integrated expression is obtained:(3)$$W=\frac{1}{2}\varepsilon_{0}\varepsilon_{T}E^{2}$$
where $\varepsilon_{0}$ and $\varepsilon_{T}$ are the vacuum permittivity and the relative permittivity of the dielectric material, respectively [1]. For example, a moderate increase of E below the breakdown field (i.e., limit for generating free charges due to bond ruptures as a consequence of material degradation) could be an alternative approach to maximize the stored energy density. Complementary, the relative permittivity of a given material could also be a useful parameter that must be considered when aiming to reach high values of stored energy. Additionally, it is highly desirable that a dielectric material displays low dielectric losses over a wide range of electric field frequencies [2,3,4,5]. Therefore, the fabrication of materials with special attention given to their specific dielectric properties represents an interesting alternative approach for obtaining high values of $\varepsilon_{T}$ (i.e., above 5). Materials (e.g., ceramics and inorganic salts) with high dielectric constants have been widely reported in the literature. However, the main drawback of these types of materials is the low breakdown field strengths that they exhibit, which represents a serious limitation to their practical application as energy storage materials. On the other hand, the use of polymer-based nanocomposites has generated growing interest aimed at exploring the possibilities of obtaining materials with high dielectric constants as well as low dielectric losses. The main reason for the synergistic combination of polymers and nanosized particles lies in the low dielectric permittivities (~2–3 at frequencies of 1 MHz) that tend to be exhibited by conventional polymers that are fabricated on a large scale, such as poly(ethylene), poly(styrene), and poly(propylene). This clearly limits the possible application of these types of polymers in capacitors or energy storage devices. Advantageously, polymeric materials display good mechanical and thermal properties, light weights, relatively easy processability, and high breakdown strengths. Thus, the incorporation of electrically conductive nanoparticles (ECNPs) dispersed in insulating polymer matrices, namely percolative nanocomposites, represents a promising approach because the ECNPs act as electrically charged domains that are spatially separated within the polymer matrix, giving rise to interfacial polarization. The above composition leads to high dielectric permittivities close to the percolation threshold for these nanocomposites. Additionally, the incorporation of small amounts of nanosized materials (e.g., graphene and carbon nanotubes) is intended to lower the percolation threshold to broaden the range of the dielectric behaviors. Mathematically, it is reasonable to infer that increasing the breakdown strength appears to be a more relevant strategy than incremental changes in the dielectric constant. Thus, an appreciable increase in electric energy storage should be expected due to *W α E*^2^. However, because the dielectric constant plays a crucial role in the energy density, low dielectric constants represent one of the significant drawbacks of pure polymer materials in energy storage applications [6]. Moreover, the presence of impurities and defects, e.g., in polymer films due to manufacturing processes, tends to notably decrease the electrical breakdown strength. Alternatively, various conductive polymers, such as poly(thiophene) and poly(aniline), have been used as fillers with relative success to obtain high dielectric constants and low dielectric losses. Interestingly, conjugated thiophene chains appear to be adequate alternatives because they exhibit a high degree of polarization due to their conjugated aromatic system. Additionally, this heterocycle is preferred over pyrrole because of the higher polarizability of sulfur atoms. Accordingly, polymers containing nanostructured thiophene domains have exhibited high permittivities and relatively low dielectric losses. This has been ascribed to the possibility of generating multiple polarized nanodipoles (<2 nm), which would enhance the dielectric properties of the polymer material. These polymer materials have displayed high dielectric constants and low dielectric losses in the 100 Hz–4 MHz range of frequencies. Thus, a variety of π-conjugated thiophene oligomer-containing polymers have been demonstrated to be potentially applicable as technological materials for energy storage capacitors.

## 2. Mechanisms of Polarization

At first, increasing the dielectric constant seems to be a highly recommended method for achieving competitive properties for electric energy storage. It is well known that polarization phenomena are the main source of the dielectric properties of a given material; therefore, an appropriate understanding of the polarization mechanisms that can predominate the dielectric properties of polymer materials is a fundamental matter. Generally, five types of polarization mechanisms can be found in different types of materials: i) interfacial, ii) ionic, iii) dipolar, iv) atomic, and v) electronic mechanisms (Figure 1).

Electronic polarization involves a distortion of the electron cloud with respect to the center of a given atomic nucleus caused by an electric field, while the atomic mechanism arises due to a relative change in the mean positions of the atomic nuclei within a molecule. However, the atomic polarization contribution is usually approximately 10–20% of the electronic polarization contribution in polymeric materials; therefore, the electronic mechanism has a significant influence on the dielectric behavior. In the case of ionic polarization, the application of an external electric field leads to small displacements of ions (>10 nm) from their equilibrium positions, giving rise to a net dipole moment. Dipolar (or orientational) polarization is ascribed to the orientation of molecular dipoles in the direction of an applied field, which predominates over their random distribution due to thermal energy. Finally, interfacial (or space-charge) polarization is related to the confined motions of charges resulting in an accumulation of charge at the interfaces of a multicomponent system (e.g., immiscible blends, organic-inorganic composites). This phenomenon creates space-charge separations under an applied electric field. Note that electronic and atomic polarizations occur at high frequencies (i.e., UV and infrared regions). In these regions, high dielectric permittivities would also be associated with significant dielectric losses; hence, the insulating properties of a material could be affected. Moreover, interfacial and ionic mechanisms would also exhibit considerable dielectric losses in the power and radio frequency ranges. Therefore, orientational polarization emerges as an attractive alternative due to of its potential for achieving high dielectric permittivity and low dielectric loss, as in the case of organic polymers. For example, much attention has been directed toward enhancing the dielectric permittivities of fluorinated polymers (e.g., poly(vinylidene fluoride), PVDF) because these materials maintain a permanent electric polarization (ferroelectric properties) that can be reversed or switched by the application of an external electric field. In this case, a reduction of the sizes of the ferroelectric regions is highly desirable because a narrowing of their hysteresis loops can be achieved. Consequently, high dielectric constants along with relatively low dielectric losses can be attained. In spite of this, fluorinated polymers exhibit high dielectric losses, which are more significant at high frequencies. This issue can be explained by the orientational polarization of the crystalline regions of these polymers, which is reflected in the broad polarization hysteresis loops (Figure 2).

## 3. Importance of the Glass Transition Temperature (T_g_) and Sub-T_g_ Transitions of Polymer Materials

The applications in energy storage of PVDF-based materials are seriously limited due to the high dielectric losses that are present. According to this strategy, efforts should be directed toward obtaining polymers with relaxor ferroelectric behavior. On the other hand, a recent approach considers the use of polymers with high glass transition temperatures that contain functional groups with high and permanent dipole moments within their chemical structure. Essentially, high T_g_ should help avoid dielectric losses that are due to the conduction phenomena relating to the electronic and ionic mechanisms. Complementary, sub-T_g_ transitions (e.g., γ and β relaxations) of dipolar entities would allow for high dielectric permittivities and low dielectric losses to be obtained. Polymers that meet these characteristics, namely, high glass transition temperatures and a considerable contribution to the dipolar polarization mechanism from sub-T_g_ transitions, are known as “dipolar glass polymers”. It must be mentioned that this concept was recently coined by Zhu et al. Note that an important requirement is that the range of temperatures of the sub-T_g_ and T_g_ transitions should be as wide as possible to ensure a high dielectric constant and that low dielectric loss is obtained, e.g., for pulse power applications. In this case, the incorporation of molecules containing functional groups with large dipolar moments is a promising alternative. The amount and size of the incorporated molecules play key roles in the subsequent dielectric behavior of the polymers. A high amount the incorporated molecule would result in excessive aggregation of the dopant molecules, causing phase separation at the micro- or nanoscale levels. Additionally, the size of these molecules should be conducive to the free orientational rotation in the region of sub-T_g_ transitions because very large dopant molecules would exhibit a steric hindrance to the free rotation of the glassy state of the polymer. Additionally, the sub-Tg transitions of these types of molecules tend to be located fairly close to T_g_. Moreover, reports in the literature indicate that the incremental changes in the dielectric constants of polymer materials using this strategy are close to 1. Because large molecules (e.g., >0.5 nm) represents a serious limitation as described above, an elegant and reasonably simple way to address these inconveniences is based on the incorporation of small attached dipolar entities as pendant groups to the polymer structure. Therefore, the presence of permanent dipolar rotational groups, mainly as side-chain dipoles, can be useful in the region of the sub-T_g_ transitions.

## 4. Small Groups with High Dipole Moments

Interestingly, side groups are more favorable to rotation, which would entail a significant increase in the dielectric permittivity along with a relatively low dielectric loss. For example, significant enhancement in the dielectric constants and relatively low dielectric losses were obtained when F, -CN, and R−S(=O)_2_−R´ groups were attached to polymers as side-chain dipoles. For example, fluorinated and cyanated poly(arylene ether nitrile) compounds, a new class of dipolar glass polymers, were obtained by an aromatic nucleophilic substitution reaction using tetramethylene sulfone (TMS) and potassium carbonate [9] (Figure 3).

A relatively high permittivity of 3.6 at 100 Hz was obtained as well as a high electrical breakdown strength of 346 MV/m and an acceptably low dissipation factor of 0.014 at 25 °C and 100 Hz. The dielectric properties were ascribed to the cooperative motions of the fluorine and cyano dipoles in the polymer chain. Additionally, the T_g_ and degradation temperatures of this polymer were approximately 180 °C and 550 °C, respectively. Treufeld et al. [10] synthesized a set of polyimides that mainly contained nitrile groups directly attached to the aromatic rings (Figure 4). The incorporation of -CN dipoles into these polymer structures resulted in an increase in their permittivities and, consequently, electrical energy storage. Notably, increasing the amount of attached nitrile groups allowed for increases in the permittivities of the polyimides. Additionally, β relaxation (i.e., a sub-T_g_ transition associated with the dipolar rotation of nitrile groups) was the main contributor to the incremental increases of the polyimide permittivities.

Figure 5 shows high-temperature dipolar polymers that were synthesized by attaching methylsulfonyl groups onto the backbone chain of commercially available poly(2,6-dimethyl-1,4-phenylene oxide) (PMSEMA) [11]. This group had a high dipolar moment (4.5 D), which contributed to achieving high dielectric constants of 6–8 below the glass transition temperature (~220 °C). Notably, depending on the degree of functionalization, an energy density, which was close to the breakdown field, of 22 J cm^−3^ and a dissipation factor of 0.003 at 1 kHz were attained. Importantly, sulfone groups incorporated into methacrylate-based polymer resulted in high dielectric constants (11.4 at 1 Hz and 10.5 at 1 kHz) and relatively low dielectric losses (tan δ∼0.02) at room temperature [11]. Interestingly, the sulfone groups also exhibited an important γ relaxation at temperatures below −100 °C and 1 Hz, considering that the T_g_ for these polymers is higher than 100 °C.

Recently, the modification of poly(epichlorohydrin) with monosulfonyl (i.e., CH_3_SO_2_-) and disulfonyl (i.e., CH_3_SO_2_(CH_2_)_3_SO_2_-) side-chain groups was performed. Thereby, high dielectric constants ranging between 7 and 11.5 in the glassy state of the polymers and low dissipation factors (tan δ ∼ 0.003−0.02) were obtained. These results indicate that the dipole density is, notably, involved in the increase in the dielectric constant. Based on these outstanding physical properties (i.e., high-temperature, high-energy-density, and low dissipation factor), these polymers have been identified as promising dielectric materials for energy storage applications [12].

Another interesting prospect emerges from the use of elastomer dielectric materials, which explores the oxidation of the thioether groups present in polysiloxanes. These functional groups are present as side groups along the backbone chain of poly(methylvinylsiloxane). The authors reported that the T_g_ of these polymers were found to be below room temperature. It was suggested that the polysiloxane backbone confers flexibility to the synthesized polymers. Additionally, depending on the presence of methylsulfone or sulfolane side groups, the T_g_ reached the maximum values of −19.2 °C or 9.3 °C, respectively. For the sulfolane group, the dielectric constants ranged between 4.9 and 22.7 at 104 Hz. Importantly, at this frequency, the lowest dielectric tangents were detected [13].

Motivated by the outstanding works of Zhu et al. in the dipolar glass polymers research field, our group has recently reported the preparation of biobased thin film nanocomposites with improved dielectric properties using-modified nanocellulose and chitosan (Figure 6), both materials are known to derive from industrial waste. The cyanoethylation of cellulose nanocrystals (CN-CNC) was achieved through a “green” method for the first time [14]. Then, the modified CNC were incorporated into a chitosan (Chi) matrix, resulting in a homogeneous and flexible material with a high dielectric constant due to the high dipole moment of the nitrile functional group. The value of the dielectric constant increases with increasing content of the modified CNC, from a value of 5.5 for pure chitosan at 25 °C and 1 kHz up to a value of 8.5 for the nanocomposite with 50 wt % CNC under the same conditions. These biobased nanocomposites show an improvement in their dielectric properties compared to those of pure chitosan and chitosan/unmodified CNC nanocomposites (for which the dielectric constant decreases to 4.5 at 25 °C and 1 kHz) and can be considered for use in high-temperature applications. Similarly, a new family of polyitaconates, containing either sulfone or nitrile side groups, were synthesized through conventional radical polymerization and their characterization and comparison with polymethacrylates containing identical groups were reported [15]. As characterized by broadband dielectric spectroscopy (BDS), all the polymers showed dielectric constants between 7 and 10 (at 25 °C and 1 kHz) and relatively low dielectric losses (≈ 0.02). The BDS measurements showed, for all the polymers analyzed, notable sub-T_g_ transitions at temperatures below −100 °C, resulting in a broad temperature interval over which these polymers exhibit high dielectric constants and can function without relatively high dielectric losses. 

## 5. Concluding Remarks and Future Directions

Dipolar glass polymers have emerged as promising versatile materials for a wide variety of energy storage applications, allowing them to simultaneously obtain relatively high dielectric constants and low dissipation factors. The high T_g_ and sub-T_g_ transitions of these types of materials contribute to these factors in important ways. Functional groups with high dipole moments (e.g., F, -CN and R−S(=O)_2_−R´) tend to result in materials with increased dielectric constants; however, the mobility effects of these groups constitute a key component in the dielectric properties. Therefore, the presence of functional groups with high dipole moments in a molecular environment with low restrictions on their mobilities may be a good combination for obtaining polymers with good dielectric properties.

Similarly, two or more functional groups with high dipole moments can be attached as side groups to the polymer structure. This allows for an increase in the density of the dipoles and, therefore, the dielectric constant of the material. On the other hand, due to the low-dissipation requirements, an increase of the temperature range between the T_g_ and sub-T_g_ transitions, in which polymers behave as a dipolar glass polymer, is highly desirable. Thus, the synthesis of polymers that contain bulky-rigid groups (e.g., norbornane groups) attached to the backbone chain, in addition to strong molecular dipoles, may be an interesting way to obtain materials with high T_g_ values, and thus a wide dipolar glass polymer temperature interval.

We believe that the remaining crucial task for future work in this area is expanding the family of potentially suitable materials that exhibit exploitable dielectric behaviors. Accordingly, a strategy incorporating highly functionalized biopolymer materials (e.g., polysaccharides) that can be chemically modified with polarizable groups may represent an interesting approach. In this context, aromatic polymers containing functional groups with high dipole moments also emerge as potential candidates. Moreover, these types of materials tend to display high T_g_ values due to the high degree of packing of the polymer chains restricting the cooperative motion of the backbone chains. Advantageously, this behavior may contribute to significantly decreasing the dielectric loss factor. On the other hand, dielectric materials that are able to combine different polarization mechanisms (e.g., dipolar and electronic) depending on the working-frequency range are candidates for potential applications in capacitor devices, such as high charge and storage electrical energy devices. Finally, broadband dielectric spectroscopy allows us to obtain valuable information on the dynamic properties (i.e., relaxation phenomena) of dielectric polymers, which is a useful way to identify and gain a better understanding of the sub-T_g_ transitions of a given material.

## Figures and Tables

**Figure 1 polymers-11-00317-f001:**
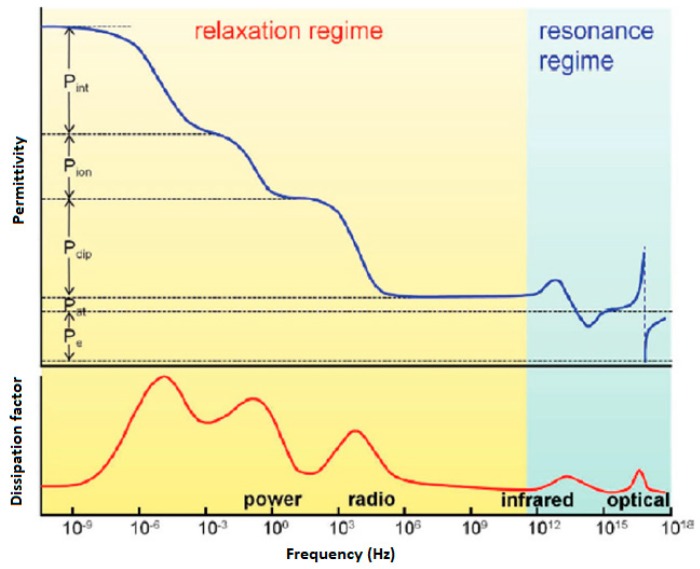
Types of polarization versus frequency in polymers. P_e_: electronic polarization; P_at_: atomic polarization; P_dip_: (dipolar) orientational polarization; P_ion_: ionic polarization and P_int_: interfacial polarization. The top panel shows the real part of the dielectric constant (ε′), and the bottom panel shows the imaginary component of the dielectric constant (dissipation factor, ε″). Reprinted with permission from reference [7]. Copyright 2012, American Chemical Society.

**Figure 2 polymers-11-00317-f002:**
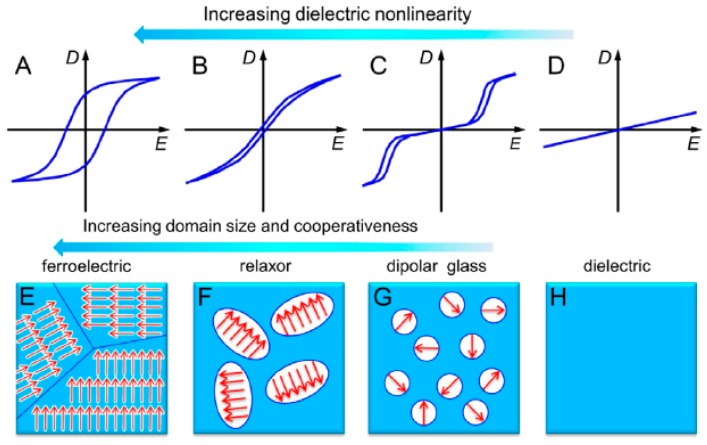
Representative illustrations of the hysteresis loops for (**A**) ferroelectric, (**B**) relaxor ferroelectric, (**C**) dipolar glass and (**D**) dielectric materials. The physical phenomena associated with these behaviors are represented in (**E**) to (**H**), respectively. Note that from (**E**) to (**H**), the polar domain size gradually decreases. Reprinted with permission from reference [8]. Copyright 2013, Elsevier.

**Figure 3 polymers-11-00317-f003:**
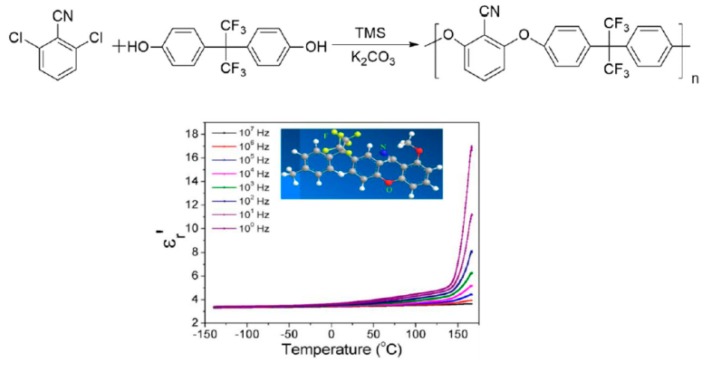
Components of the dielectric constant (ε′_r_) as a function of temperature at different frequencies for fluorinated and cyanated poly(arylene ether nitrile). Adapted from reference [9] with permission. Copyright 2018, Wiley.

**Figure 4 polymers-11-00317-f004:**
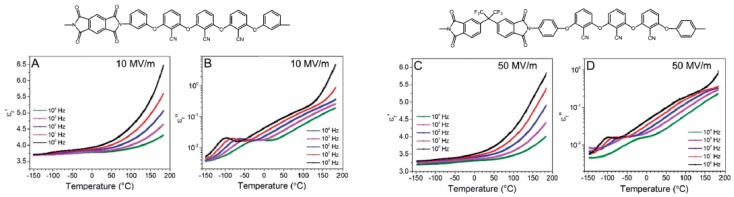
Dependence of the real part of the relative permittivity (ε′) and dielectric loss (ε″) on the temperature for poly(amide) samples under a high-strength electric field with peak amplitudes of 10 MV m^−1^ and 50 MV m^−1^. Adapted from reference [10] with permission. Copyright 2014, Royal Society of Chemistry.

**Figure 5 polymers-11-00317-f005:**
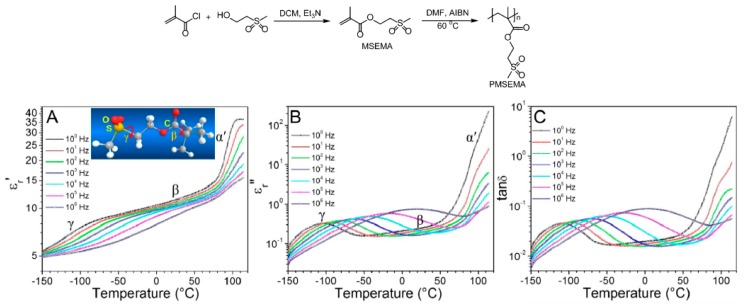
(**A**) Real (ε′_r_) and (**B**) imaginary (ε″_r_) parts of the relative permittivities and (**C**) dissipation factors (tan δ) as a function of temperature at different frequencies for PMSEMA. Adapted from reference [11] with permission. Copyright 2015, American Chemical Society.

**Figure 6 polymers-11-00317-f006:**
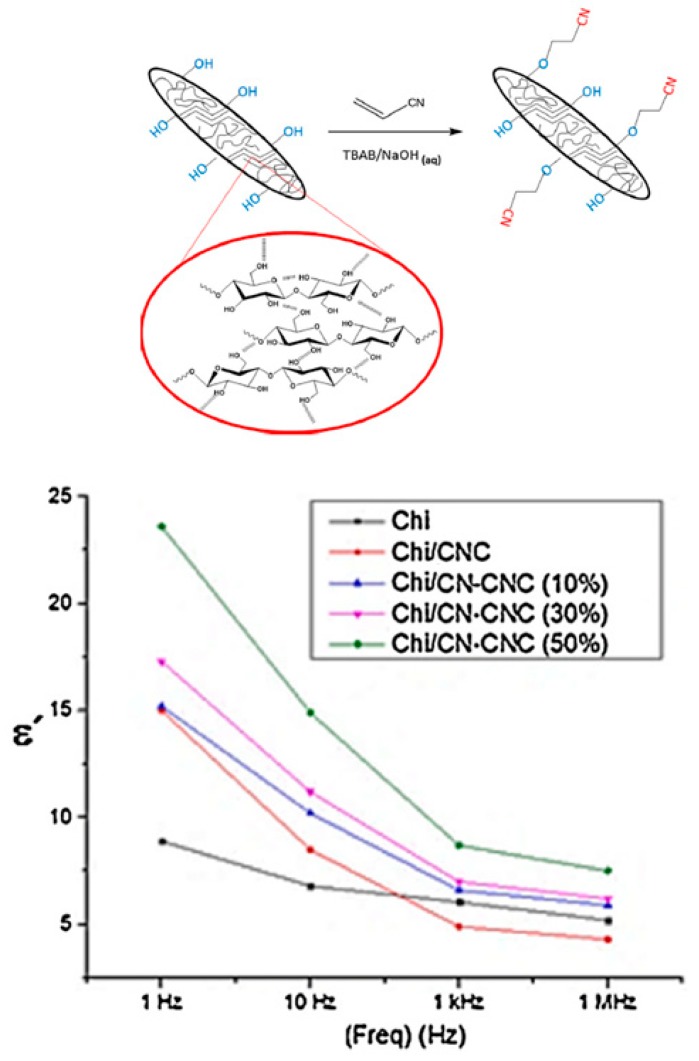
ε′ values at different frequencies for chitosan, the nanocomposite with unmodified CNCs and those with different amounts of cyanoethylation of cellulose nanocrystals (CN-CNC). Adapted from reference [14] with permission. Copyright 2018, Elsevier.

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
