# Peer review of "Dipolar Glass Polymers Containing Polarizable Groups as Dielectric Materials for Energy Storage Applications. A Minireview"

_polymers, 2019, doi:10.3390/polym11020317_

Round 1
Reviewer 1 Report
The dielectric properties of materials determine whether they are suitable for the application of energy storage. The preparation of polymeric materials with both high dielectric constant and low dielectric loss is the key and a challenge. Facing this challenge, this work fully discussed a candidate promising approach by modifying or attaching side groups with high dipolar moments on the backbone chain of polymer along with summarizing the latest related works. The text is logical and well organized. As a mini review, the abstract is slightly long and advice to make it shorter and more concise. The authors missed out a few closely relevant literatures published most recently. In addition, the authors should cite literatures to support their analysis in appropriate positions throughout the manuscript, such as the sentences in line 160 of page 5 and line 149 of page 4. Overall, this work may be of interests in a wide audience. Thereore, I recommend it to be accepted for publication in polymers after minor resions.
Author Response
The dielectric properties of materials determine whether they are suitable for the application of energy storage. The preparation of polymeric materials with both high dielectric constant and low dielectric loss is the key and a challenge. Facing this challenge, this work fully discussed a candidate promising approach by modifying or attaching side groups with high dipolar moments on the backbone chain of polymer along with summarizing the latest related works. The text is logical and well organized. As a mini review, the abstract is slightly long and advice to make it shorter and more concise. The authors missed out a few closely relevant literatures published most recently. In addition, the authors should cite literatures to support their analysis in appropriate positions throughout the manuscript, such as the sentences in line 160 of page 5 and line 149 of page 4. Overall, this work may be of interests in a wide audience. Thereore, I recommend it to be accepted for publication in polymers after minor resions.
Dear reviewer, many thanks for your appreciable contributions and suggestions. We are very grateful for your comments. So considering your main concerns about this manuscript, the responses and/or comments are given point by point below.
- Considering your suggestions, the abstract length was shortened. Additionally, the title of the manuscript was also modified, in order to make it more evident and concise for the readers.
- More literature citations were added to support the introduction section of the manuscript.
Reviewer 2 Report
Manuscript Number: polymers-437261 by Bonardd et al. and titled: Dipolar Glass Polymers Containing Polarizable Groups as Dielectric Materials for Energy Storage Applications: A Minireview.
This is a manuscript based on 15 references. This manuscript can be published in Polymers after a major revision. Below are the reviewer’s comments.
1. The authors need Change the title (i.e. energy storage devices to capacitors)? Can these dielectric materials (Dipolar Glass Polymers Containing Polarizable Groups) be used in Lithium ion batteries? Cite reference on the use of dielectric materials in LIBs. Good references on polymer nanofibers separators and ceramic/carbon composite fibers for LIBs: (Electrochimica Acta, 224, PP. 608-621, (2017), Journal of Alloys and Compounds: Volume 686, Page 733-743, (2016); and Journal of Applied Polymer Science: Volume 133 (2016), Article Number: 42847). In fact, Energy storage materials are not necessarily used as capacitors? mention different type of energy storage (batteries, supercapacitors, capacitors and fuel cells.
2. The authors provided a nice introduction, but it lacks some discussion on nanocomposites and nanofibers as nanodielectrics in conventional capacitors and supercapacitors. See work by Nelson JK et al (Nanocomposite dielectrics - properties and implications), See work by Tanaka T et al (IEEE Transactions on Dielectrics and Electrical Insulation; Vol. 12, No. 5; October 2005] and work by Lewis: Interfaces: nanometric dielectrics; J. Phys. D: Appl. Phys. 38 (2005) 202–212; doi:10.1088/0022-3727/38/2/004]).
3. Results on the use of polymer composite nanofibers as dielectric materials need to be briefly discussed and cited in the introduction? See work published on nanofibers (Nelson JK and coworkers.: Effect of High Aspect Ratio Filler on Dielectric Properties of Polymer Composites: A Study on Barium Titanate Fibers)
4. The authors need to discuss recent results on the breakdown of nanocomposites and composite nanofibers and on the use of Dipolar Glass Polymers as higher breakdown strength materials (refs are available in the literature)
5. Please distinguish the difference between permittivity and dielectric constant. Define dielectric constant in the intro
6. This manuscript contains a few technically flawed claims and English errors and must be revised and proofread by English language expert.
For example, :
In the abstract: capacitor devices?
Line#44: Qis?
Line#71:The above composition?
Line#154:A high amount the incorporated molecule?
Author Response
Reviewer #2
Manuscript Number: polymers-437261 by Bonardd et al. and titled: Dipolar Glass Polymers Containing Polarizable Groups as Dielectric Materials for Energy Storage Applications: A Minireview.
This is a manuscript based on 15 references. This manuscript can be published in Polymers after a major revision. Below are the reviewer’s comments.
Dear reviewer, many thanks for your appreciable contributions and suggestions. We are very grateful for your comments. So considering your main concerns about this manuscript, the responses and/or comments are given point by point below.
1. The authors need Change the title (i.e. energy storage devices to capacitors)? Can these dielectric materials (Dipolar Glass Polymers Containing Polarizable Groups) be used in Lithium ion batteries? Cite reference on the use of dielectric materials in LIBs. Good references on polymer nanofibers separators and ceramic/carbon composite fibers for LIBs: (Electrochimica Acta, 224, PP. 608-621, (2017), Journal of Alloys and Compounds: Volume 686, Page 733-743, (2016); and Journal of Applied Polymer Science: Volume 133 (2016), Article Number: 42847). In fact, Energy storage materials are not necessarily used as capacitors? mention different type of energy storage (batteries, supercapacitors, capacitors and fuel cells.
2. The authors provided a nice introduction, but it lacks some discussion on nanocomposites and nanofibers as nanodielectrics in conventional capacitors and supercapacitors. See work by Nelson JK et al (Nanocomposite dielectrics - properties and implications), See work by Tanaka T et al (IEEE Transactions on Dielectrics and Electrical Insulation; Vol. 12, No. 5; October 2005] and work by Lewis: Interfaces: nanometric dielectrics; J. Phys. D: Appl. Phys. 38 (2005) 202–212; doi:10.1088/0022-3727/38/2/004]).
3. Results on the use of polymer composite nanofibers as dielectric materials need to be briefly discussed and cited in the introduction? See work published on nanofibers (Nelson JK and coworkers.: Effect of High Aspect Ratio Filler on Dielectric Properties of Polymer Composites: A Study on Barium Titanate Fibers)
4. The authors need to discuss recent results on the breakdown of nanocomposites and composite nanofibers and on the use of Dipolar Glass Polymers as higher breakdown strength materials (refs are available in the literature).
(This response corresponds to points 1-4)
You are right; the title of the manuscript was modified as you suggest, in order to make it more evident and concise for the readers. The suggested cites were incorporated in the text of the manuscript. Additionally, in the introduction other applications are mentioned, for example, energy storage, digital memory devices, pulsed power systems, and signal processing. According to the literature, these fields of applications are concordant with the properties of dielectric polymer materials. Note that the literature citations suggested by you were added to support the introduction section of the manuscript.
On the other hand, we think the inclusion of nanocomposites (i.e., nanofibers, nanometrics dielectrics) could divert the focus of this minireview based on the potential use of all-polymer materials as dielectrics (e.g., for more clarity the title of the mini review was changed). In the text of the manuscript some examples of nanomaterials are mentioned and cited, in order to give an adequate context to all-polymers in the research field of the dielectrics. Considering this, the state of art of the development of nanomaterials for energy storage applications, by themselves would deserve a special review apart. Because of the properties and phenomena of the nanoscale materials could exhibit relevant effects and play a key role in the dielectric characteristics of a specific material.
5. Please distinguish the difference between permittivity and dielectric constant. Define dielectric constant in the intro
You are right: a brief definition was introduced in the abstract (marked in red)
6. This manuscript contains a few technically flawed claims and English errors and must be revised and proofread by English language expert. For example : In the abstract: capacitor devices? Line#44: Qis? Line#71:The above composition? Line#154:A high amount the incorporated molecule?
The manuscript was revised by Wiley Editing Services to avoid spelling errors, typos and poor writing. Additionally, the mentioned examples were also corrected (marked in red)
Round 2
Reviewer 2 Report
The reviewer is satisfied with the corrections and answers from the authors and agrees to publish this paper in polymers after revising the title /see below/
Please revise the title by changing the words All-Polymers to Polymeric. The revise title should be:
Polymeric.dielectric materials for energy storage applications. A minireview.